# Seasonal Variation Characteristics and the Factors Affecting Plankton Community Structure in the Yitong River, China

**DOI:** 10.3390/ijerph192417030

**Published:** 2022-12-18

**Authors:** Ang Dong, Xiangfei Yu, Yong Yin, Ke Zhao

**Affiliations:** 1Key Laboratory of Songliao Aquatic Environment, Ministry of Education, School of Municipal and Environmental Engineering, Jilin Jianzhu University, 5088 Xincheng Street, Changchun 130118, China; 2Changchun Municipal Engineering & Research Institute Co., Ltd., Changchun 130022, China

**Keywords:** zooplankton, phytoplankton, Cyanophyta, Yitong River, seasonal variation, plankton community

## Abstract

To explore how environmental factors affected the plankton structure in the Yitong River, we surveyed the water environmental factors and plankton population in different seasons. The results showed high total nitrogen concentrations in Yitong River throughout the year, while the total phosphorus, water temperature (WT), and chemical oxygen demand in summer were significantly higher than those in other seasons (*p* < 0.05), and the dissolved oxygen (DO) concentrations and TN/TP ratio were significantly lower (*p* < 0.01) than those in other seasons. There was no significant seasonal change in other environmental factors. Cyanophyta, Chlorophyta, and Bacillariophyta were the main phytoplankton phylum, while Protozoa and Rotifera were the main zooplankton phylum. The abundance and biomass of zooplankton and phytoplankton in the summer were higher than those in other seasons. Non-Metric Multidimensional scaling methods demonstrated obvious seasonal variation of phytoplankton in summer compared to spring and winter, while the seasonal variation of the zooplankton community was not obvious. The results of the redundancy analysis showed that WT, DO and nitrate nitrogen were the main environmental factors affecting phytoplankton abundance. In contrast to environmental factors, phytoplankton was the main factor driving the seasonal variation of the zooplankton community structure. Cyanophyta were positively correlated with the changes in the plankton community.

## 1. Introduction

Plankton, comprising phytoplankton and zooplankton, is an ecologically important group in aquatic ecosystems [1]. As primary producers in the food chain, phytoplankton are important in the material cycle and energy transfer process of the aquatic ecosystem [2], due to their short growth cycle and rapid response to aquatic environmental changes, thus enabling its community structure characteristics to directly reflect the changes of aquatic environment [3,4]. As primary consumers in aquatic ecosystems, zooplankton are important in transferring energy from lower to higher organisms, thereby regulating the growth of phytoplankton [5], whose community structure characteristics are mostly regulated by the composition and abundance of phytoplankton and their predators [6]. Plankton community and environmental factors are closely correlated, with the environmental factors directly or indirectly affecting the temporal variation characteristics of the plankton community structure. Furthermore, the species composition and distribution of the plankton community can objectively reflect the changing patterns of the water environment [7,8]. Therefore, the succession patterns of plankton communities are an important basis to reveal the changes of the aquatic ecosystem. The response of the plankton community structure to environmental factors can reflect the health status of the river ecosystem and provide guidance to local river management [9]. As an important primary producer in the aquatic ecosystem, Cyanophyta are prone to blooms in eutrophic rivers, and the resulting toxicity affects the plankton community structure and species diversity via the food web [10]. Previous studies have explored the main influencing factors of phytoplankton [11,12] or zooplankton [13,14], such as total nitrogen, total phosphorous and water temperature. However, there are few reports on the correlation between the overall community of both and environmental factors and the Cyanophyta community structure in urban rivers. In this context, it is vital to explore the temporal dynamics of the zooplankton and phytoplankton community structure and its influencing factors for maintaining the balance of the aquatic ecosystem.

The Yitong River is one of the largest secondary tributaries of the Songhua River, China. Due to its small natural water volume and weak self-purification capacity, its water flow is greatly affected by precipitation and the season. In the up and down stream of the river, which are mostly rural and suburban areas, crops are grown along the river and there is a risk of high nitrogen and phosphorus content seeping from fertilizer use, thus leading to eutrophication of the water body. The Yitong River as an inland urban river passes through the main urban zone of Changchun. With the rapid development of urbanization, an increasing number of treated domestic sewage discharged into the river resulting in its inferior Class V surface water quality (GB 3838-2002) [15]. Ji et al. [16] analyzed the black-odor mechanism of the Yitong River and found that organic matter was the main source of pollution in the Yitong River and its tributaries, and most of the watershed was contaminated with heavy metals to varying degrees. Human activities and production facilities were the main sources of antibiotic pollution in the Yitong River [17]. Previous studies have explored the correlation between phytoplankton and environmental factors in the urban section of the Yitong River, but little research has focused on the factors affecting the plankton community structure of the overall watershed scale [18]. Recently, local government has focused on the Yitong River’s water environment quality and issued multiple policies to solve water environmental pollution. Since then, the water environment quality has significantly improved and the water ecosystem has gradually recovered. However, while the Yitong River, as an important urban water body of Changchun city and the routine water quality monitoring, does not accurately reflect its complex water environment quality [19], there is an urgent need to explore the seasonal variation characteristics and influencing factors of the plankton community structure in the Yitong River basin, and to objectively evaluate the status quo of its water environmental quality and the change in trend on the temporal scale.

With accelerating urbanization, there is a close connection between aquatic ecosystems and human activities, especially in urban rivers [20]. As a reliable indicator of aquatic ecosystem deterioration, plankton is essential for the health maintenance of urban rivers by revealing its community dynamics and influencing factors. In this research, the Yitong River was selected as the research object. The characteristics of environmental factors and plankton community were analyzed seasonally. The dynamics of the plankton community structure and its corresponding relationship with environmental factors were explored through Non-Metric Multidimensional scaling methods (NMDS) and Redundancy Analysis (RDA). The impacts of Cyanophyta on the plankton community were assessed by Pearson’s correlation analysis. Our study aims to (1) clarify the characteristics of seasonal variations in the plankton community structure of the Yitong River; and (2) reveal the main environmental factors influencing the variations of the plankton community. This research can provide a theoretical basis for the health maintenance of the inland urban river ecosystem.

## 2. Materials and Methods

### 2.1. Study Area

The Yitong River originates in the Yitong Manchu Autonomous County of Jilin Province, China. It flows through downtown Changchun in its upper and middle reaches and finally flows into the Second Songhua River. Its total length is 343.5 km, which covers an area of 7515 km^2^. There were 7 sampling points (Figure 1) from up stream to down stream, which were the Xingguang section (S1), Xinlicheng Reservoir dam section (S2), in front of the south gate of the South Third Ring Road (S3), in front of the Free Gate (S4), Yangjiawaizi Bridge section (S5), Baolong Bridge section (S6), and Kaoshan Bridge section (S7) (Table 1). Sampling points S1, S2, S5, S6, and S7 were water quality monitoring points controlled by the state and the province. Upstream point S1 and downstream point S6 and S7 are in rural areas, which are susceptible to interference from different human factors, like farmland irrigation and domestic sewage. As an inland urban river, the Yitong River passes through the urban area of Changchun, and the monitoring work of sampling points S2–S5 can completely reveal the water environmental quality of the Yitong River from top to bottom in Changchun.

### 2.2. Sampling and Analysis

Sample collection from the Yitong River was carried out in 2021 during different seasons, namely spring (April), summer (June and August), autumn (October) and winter (December), with a sampling frequency of once a month. Dissolved oxygen (DO), pH and water temperature (WT) were measured directly using the WTW Multi3320 (Xylem Analytics Germany Sales GmbH & Co. KG, WTW, Weilheim, Germany) and INESA JPBJ-608 portable dissolved oxygen tester (Shanghai INESA Scientific Instrument Co., Ltd., Shanghai, China). Water transparency (SD) was determined by the Secchi disk. Water samples for Plankton (21 L), environmental factors (2 L) and Chlorophyll-a (Chl-a) (1 L) analysis were all collected at 0.5 m water depth at each sampling site and stored at 0–4 °C. Total nitrogen (TN), ammonium nitrogen (NH_4_^+^-N), nitrate nitrogen (NO_3_^−^-N), total phosphorus (TP), chemical oxygen demand (COD_Cr_), permanganate index (COD_Mn_) and Chl-a were determined in the laboratory within 24 h according to the Water and waste water monitoring and analysis method (fourth edition) [21]. The water flow data (Q) of the Yitong River in different seasons were obtained from the National Water Information Network “http://xxfb.mwr.cn/ (accessed on 10 December 2022)”.

Water samples for Phytoplankton, Rotifera and Protozoa analysis were collected in 1 L polyethylene bottles and immediately fixed by adding Lugol’s solution and 4% formaldehyde solution. They were left standing in the laboratory, and after 48 h the clear liquid was siphoned off to concentrate the sample from 1 L to 30 mL. Water samples for Cladocera and Copepoda analysis (20 L) were filtered by 64 µm mesh size net and condensed to 100 mL, and then 4% formaldehyde solution was immediately added to fix. Concentrated samples of phytoplankton and zooplankton were identified and counted under the Olympus CX23 microscope (Olympus Lifescience, Shinjuku, Tokyo, Japan) using 0.1 mL and 1 mL counting frames, respectively. Phytoplankton and zooplankton species were identified by using manuals [22,23]. Zooplankton biomass was measured via the volumetric method. The volume of zooplankton was then converted to biomass based on the unit weight of the different species [24]. The zooplankton biomass was expressed in wet weight.

### 2.3. Data Processing and Analysis

The trophic level index (TLI) was determined by five environmental indicators, including Chl-a, SD, TP, TN, and COD_Mn_, to assess the trophic state of the Yitong River [25]. The computational formula for TLI of a single indicator and comprehensive TLI are:(1)TLI(Chl-a)=10(2.5+1.086ln(Chl-a))
(2)TLI(TP)=10(9.436+1.624ln(TP))
(3)TLI(TN)=10(5.453+1.694ln(TN))
(4)TLI(SD)=10(5.118−1.94ln(SD))
(5)$$\mathrm{TLI}\left({\mathrm{COD}}_{\mathrm{Mn}}\right)=10(0.109+2.661\ln\left({\mathrm{COD}}_{\mathrm{Mn}}\right))$$
(6)$$\mathrm{TLI}\left(\Sigma \right)=\sum^{\;}W_{j}\times \mathrm{TLI}(j)$$
where the unit of Chl-a is mg/m^3^, the unit of SD is m, while the others are mg/L. TLI (Σ) is the comprehensive nutritional status index; W_j_ is the weight of the nutritional status index of the jth parameter; TLI (j) is the nutritional status index of the jth parameter. TLI ≤ 30 represents oligotrophic, 30 < TLI ≤ 50 represents mesotrophic, 50 < TLI ≤ 60 represents light eutrophic, 60 < TLI ≤ 70 represents middle eutrophic, TLI > 70 represents highly eutrophic.

Shannon-Wiener index (H’) [26] and Pielou index (J) [27] were used to describe the characteristics of plankton biodiversity, and the computational formulas were as follows:(7)$$H^{\prime }=-\overset{s}{\underset{i=1}{\sum }}(\frac{n_{i}}{N})\log_{2}(\frac{n_{i}}{N})$$
(8)$${J=H}^{\prime }{/\log}_{2}S$$
where n_i_ is the number of individual species I; N is the total number of individuals of all samples; S is the total number of plankton species in the samples.

SPSS 26.0 was used for one-way analysis of variance (ANOVA). The LSD method was used to test the differences between physical and chemical indexes of various water bodies and the plankton community structure between seasons (homogeneity of variance test and normality test were performed on data before variance analysis). Pearson’s correlation analysis was used to explore the correlation between the Cyanophyta community structure characteristics and plankton biodiversity index, phytoplankton and zooplankton. The *p*-value represented the significance between the two variables, and generally requires *p* < 0.05 to be statistically significant. NMDS was used to explore the seasonal dynamics of the plankton community, while the analysis was completed by vegan package, and the NMDS was implemented by metaMDS function in R-4.1.2. Correlation heat map and linear fit relationship plots were generated via the corrplot package. RDA was performed using the CANOCO 5.0 software to explore the relationship between plankton communities and environmental factors. All data were entered in the lg (x + 1) format, besides the pH before analysis to achieve a normal distribution. Correlation charts were implemented in Origin 2021 and R-4.1.2 software, and the drawing of sampling points was completed in ArcGis 10.6.

## 3. Results

### 3.1. Seasonal Variation of Environmental Factors

Figure 2 shows the seasonal variation trend of environmental factors of the Yitong River in 2021. During the study period, the monthly average TN concentration varied between 3.25 ± 1.82 and 4.78 ± 2.40 mg/L, and there was no significant difference between seasons (ANOVA, *p* > 0.05). NO_3_^−^-N was the main pollutant of N source nutrient in Yitong River compared to NH_4_^+^-N. The Yitong River TP concentration (0.31 ± 0.07 mg/L) in summer was significantly higher than those in other seasons (ANOVA, *p* < 0.05), while TN/TP ratio (N/P) in summer was significantly lower than those in other seasons (ANOVA, *p* < 0.01). The summer COD_Cr_ concentration (43.16 ± 17.25 mg/L) was significantly higher than those in spring and winter (ANOVA, *p* < 0.05), whereas the DO concentration was significantly lower than those in spring and winter (ANOVA, *p* < 0.01), thereby showing good synchronization with the indexes of nutrient and organic pollution degree. Except spring and autumn, WT differences in other seasons were significant (ANOVA, *p* < 0.01), with annual variations ranging from 1.29 ± 0.88 to 22.11 ± 2.34 °C. The annual pH range of Yitong River was 7.69 ± 0.54–8.29 ± 0.52, indicating an alkaline water body. There were no significant seasonal differences in pH, COD_Mn_, and SD during the study period (ANOVA, *p* > 0.05). From the TLI calculation results, we can conclude that the Yitong River was highly eutrophic in spring, summer, and autumn, while being middle eutrophic in winter, and there were significant differences between the TLI values in summer and winter (ANOVA, *p* < 0.05).

### 3.2. Seasonal Variation of Plankton Community Abundance, Biomass and Biodiversity Indexes

We identified 168 species of phytoplankton from nine phyla during the study. Cyanophyta, Chlorophyta, and Bacillariophyta dominated the annual species composition, accounting for over 90% of phytoplankton abundance in each season (Figure 3a). In terms of season, Bacillariophyta dominated in spring, accounting for 81.76% of the total phytoplankton abundance. The average abundance of phytoplankton peaked in summer (54.17 × 10^6^ cells/L), with Chlorophyta and Cyanophyta being the dominant species, accounting for 41.78% and 33.17% of the total phytoplankton, respectively. The average phytoplankton abundance decreased from autumn to winter, while decreasing significantly in winter as compared with summer (ANOVA, *p* < 0.05). Cyanophyta and Bacillariophyta dominated in autumn, accounting for 38% and 33.62% of total phytoplankton, respectively, whereas Bacillariophyta dominated in winter, accounting for 73.14%.

We identified 93 species of zooplankton from four phyla, with Protozoa and Rotifera dominated in abundance (Figure 3b). In terms of season, the abundance of Protozoa in spring, summer, autumn and winter was 280 ind/L, 221.43 ind/L, 76.43 ind/L and 148.57 ind/L, accounted for 47.37%, 11.11%, 13.65%, and 57.86%, respectively; the abundance of Rotifera was 304.29 ind/L, 1749.64 ind/L, 479.29 ind/L and 102.86 ind/L, accounted for 51.48%, 87.78%, 85.59%, and 40.06%, respectively; Copepoda and Cladocera accounted for 1% and less than 1%. The total zooplankton abundance in summer (1993.30 ± 727.57 ind/L) was significantly higher than those in autumn (560 ± 198.22 ind/L) and winter (256.76 ± 63.64 ind/L) (ANOVA, *p* < 0.05). 

In terms of biomass, Rotifera dominated the annual biomass of zooplankton with 0.37 mg/L (spring), 2.10 mg/L (summer), 0.58 mg/L (autumn) and 0.12 mg/L (winter), accounting for 90.24% (spring), 95.89% (summer), 93.55% (autumn), and 80.84% (winter), respectively. Phytoplankton biomass (expressed as Chl-a) showed the same temporal trend as zooplankton biomass, with the peak in summer and the trough in winter. In comparison to autumn and winter, the biomass was significantly higher in the summer (2.19 ± 0.90 mg/L) (ANOVA, *p* < 0.01) (Figure 3c). The correlation results showed that Chl-a concentrations showed significant positive correlations with all zooplankton biomass, except for Cladocera (Pearson, *p* < 0.01) (Figure 3f).

The Shannon-Wiener index (H’) and Pielou index (J) of phytoplankton in the Yitong River during spring were significantly lower than those in summer and autumn (ANOVA, *p* < 0.05), and the H’ of phytoplankton in spring was significantly lower than that in winter (ANOVA, *p* < 0.05), with no significant difference among zooplankton (ANOVA, *p* > 0.05) (Figure 3d,e).

### 3.3. Dynamic Changes of Plankton and Its Relationship with Environmental Factors

In Figure 4, NMDS analysis showed that the phytoplankton abundance at the phylum level during summer differed from those in spring and winter. Furthermore, the seasonal variation of zooplankton abundance at the phylum level was similar, with a high coincidence rate being observed in ellipses, and the zooplankton community distribution in autumn was more dispersed than those in other seasons.

To further explore the correlation between plankton community and environmental factors, we found that the first axial lengths of phytoplankton and zooplankton were 1.98 and 1.97, respectively, by using the Detrended Corresponding Analysis (DCA), and the gradient length was less than four. Therefore, RDA was used to more accurately explain the main driving environmental factors throughout the year in the Yitong River Basin. Both the physical and chemical factors were tested by the Monte Carlo simulations with 499 permutations, and we subsequently screened the environmental factors (*p* < 0.05) (Figure 5). In terms of phytoplankton, the main environmental factors affecting the abundance of the phytoplankton community were WT (*p* = 0.002), DO (*p* = 0.004), and NO_3_^−^-N (*p* = 0.014), and the total contribution rate of the three indexes was as high as 74.8%. For zooplankton, DO (*p* = 0.01) was the main environmental factor affecting the abundance of the zooplankton community, but Chl-a, an important indicator of phytoplankton biomass, had a more significant effect on the zooplankton community (*p* < 0.01), with a total contribution rate of 61.5% from both (Table 2).

### 3.4. Correlation between Cyanophyta and the Plankton Community

The spatial and temporal distribution of Cyanophyta abundance in the Yitong River is shown in Table 3. The linear fits and correlation analysis of Cyanophyta abundance to plankton biomass and diversity index were carried out. The results showed that the Cyanophyta abundance had highly significant positive correlations with the biomass of both phytoplankton and zooplankton (Pearson, *p* < 0.01) (Figure 6a,b) and Shannon-Wiener index for phytoplankton (Pearson, *p* < 0.05) (Figure 6c).

## 4. Discussion

### 4.1. Effect of the Environmental Factors on Seasonal Changes of Phytoplankton

RDA results showed that the main environmental factors affecting the phytoplankton community characteristics in the Yitong River were WT, DO and NO_3_^−^-N. WT showed a significant positive correlation with most phytoplankton, especially Cyanophyta (Figure 5a), due to several reasons. First, light and WT are two important environmental factors for the growth of phytoplankton [28]. Since these two factors peak in summer, phytoplankton grows the fastest in summer, and Cyanophyta especially prefers high WT, making the WT of the Yitong River suitable for Cyanophyta growth in summer. In winter and spring (Figure 4a), the river freezes and thaws, respectively, and most phytoplankton either cannot survive or exist as spores due to the low temperatures [29]. However, Bacillariophyta, due to its higher surface-area-to-volume ratio, can adapt better to the low-light environment in the water body during winter season [30]. In addition, due to its high Chl-a content per unit volume [31], even in the low light and low temperature conditions, Bacillariophyta can still active photosynthesize, which increases the DO concentration, thus indicating its good adaptability to different environments. This explains the Bacillariophyta seasonal difference being insignificant, since the WT and Bacillariophyta correlation is not high, but the DO concentration was positively correlated with Bacillariophyta. It is also possible that higher temperatures reduce the viscosity of surface water, which accelerates the diffusion of nutrients to the cell surface [32]. This decrease in viscosity also promotes the sinking of larger, non-mobile Bacillariophyta, allowing both Cyanophyta and Chlorophyta to dominate during the summer. DO was negatively correlated with Cyanophyta and Chlorophyta, except during winter when the Yitong River was highly eutrophic, and the excessive growth and death of phytoplankton decreased the DO concentration in the river [33]. For example, from spring to summer, the degree of eutrophication of the river increased, resulting in the phytoplankton species composition in the Yitong River changing from Bacillariophyta to Cyanophyta and Chlorophyta, accompanied by a sharp drop in the DO concentration (Figure 2). Secondly, the growth of phytoplankton increased Chl-a levels in summer, thereby decreasing the river transparency and reducing the access to light necessary for phytoplankton growth [34] and increasing the phytoplankton respiration more than photosynthesis, thus finally leading to the decrease of DO concentration in the river. In addition, the increase of WT improved the concentration of ions in the river, thus reducing the solubility of oxygen, which may be another reason for the decrease of DO concentration in summer [35].

Nitrogen and phosphorus, being basic nutrients, are vital for the phytoplankton growth. The results of multiple studies showed that the phytoplankton community structure was mostly affected by nitrogen and phosphorus [12,36,37,38]. According to the ratio of Redfield [39], N/P = 16 is the most suitable atomic ratio of cell composition for phytoplankton growth. If the N/P ratio exceeds 16:1, phosphorus is considered as a limiting factor, and vice versa. Our results also obeyed this theory. In summer, when N/P ratio was close to 16:1 (Figure 2), the abundance and biomass of phytoplankton reached its peak value (Figure 3a,c). The results of the RDA demonstrated a significant negative correlation between NO_3_^−^-N and phytoplankton, indicating that NO_3_^−^-N, as the main nutrient in the Yitong River, caused the seasonal variation of N/P ratio and had a greater influence on phytoplankton growth than TN. However, the Yitong River was N overloaded, phytoplankton in water cannot fully uptake nutrients for their own growth in other seasons, resulting in the significant negative correlations between NO_3_^−^-N and phytoplankton.

The results of the study was similar to those of northern Chinese rivers, such as the Genhe River in the Greater Hinggan Mountains [40] and landscape waters in Tianjin city [41], in that WT had a significant effect on the dynamics of phytoplankton communities. Meanwhile, the results of the study were consistent with the coastal waters of Qinhuangdao, China [42] and the Muling River [36], where both WT and DO were the main environmental factors affecting phytoplankton. Phytoplankton in other regional watersheds were affected by nutrients to different degrees, and Rao et al. [43] found that the effect of nutrients on phytoplankton was higher in Lake Luhu than in WT. However, in our study, the phytoplankton in the Yitong River basin was more significantly affected by WT due to phosphorus limitation.

### 4.2. Effect of the Environmental Factors on Seasonal Changes of Zooplankton

We explored the seasonal variation of zooplankton abundance to reveal their response to changes in the water environment of the Yitong River. According to the NMDS results, the confidence intervals of zooplankton in different seasons mostly overlapped (Figure 4b), which indicated that the zooplankton community characteristics was similar in terms of expression at different temporal scales. The annual species composition of zooplankton in the river habitats is dominated by Rotifera, since they grow fast and have good adaptability to changes in physical and chemical environments as well as hydrological conditions in rivers [44,45]. With reference to earlier studies [46,47,48], Chl-a was applied to the RDA as a significant indicator for describing phytoplankton biomass [38,49] and a potential influence on zooplankton. The results showed that there was a positive correlation between Chl-a and four zooplankton phyla, i.e., Figure 5b, the Chl-a concentration significantly promoted the abundance of zooplankton in the river, which also indirectly indicated that there might be a good predation relationship between zooplankton and phytoplankton in the Yitong River basin. The seasonal dynamics of zooplankton were more influenced by phytoplankton than by other environmental factors (Table 2). Besides, being affected by biological factors like food supply, competition and predation, zooplankton community characteristics are also closely related to abiotic factors. RDA demonstrated that DO was closely correlated with the zooplankton community structure. DO is necessary for the survival of organisms and conducive to growth and reproduction of zooplankton through respiration [50]. The results of the study showed obvious similarity with other regions in that DO was the main environmental factor affecting zooplankton variations [51,52]. Unlike other regions where WT is an important environmental factor affecting the community structures of zooplankton [53,54]. However, in this study, the effect of WT on zooplankton was not significant, and it was more likely to indirectly affect the dynamic changes in the zooplankton community via altering the structure of the phytoplankton community.

### 4.3. Response of Zooplankton to Variation in Dynamic Phytoplankton Community Structures

The RDA results demonstrated that the seasonal variation of zooplankton in the Yitong River corresponded more significantly to biological factor (Chl-a) than to environmental factors. To further explore the response of zooplankton to the dynamic changes of phytoplankton community structure, the annual variation trends of both zooplankton and phytoplankton biomass were compared and we found that the two trends showed obvious synchronicity with seasonal changes (Figure 3c). Higher WTs and optimum nutrient concentrations in summer were important reasons for the rapid growth of phytoplankton biomass, which simultaneously affected the growth trend of zooplankton. Since Chrysophyta, Cryptophyta, and Dinophyta use flagella to capture nutrients for their growth and reproduction and effectively avoid predators, they appeared in all seasons [55] (Figure 3a). Pearson’s correlation analysis results showed that there was a significant positive correlation between the zooplankton and phytoplankton biomass (Figure 3f), which further proved our point. Rotifera accounted for the largest proportion in the annual zooplankton biomass, and the positive correlation between Rotifera and phytoplankton biomass was due to predation relations between rotifers and algae, bacteria, and organic detritus [56]. This happens even during the colder spring and winter periods, since there is organic detritus left in the river over from the death of aquatic organisms in the previous year, for Rotifera to prey on. Since the other predators are less active during this period, Rotifera remains the largest share of zooplankton biomass, even during cooler seasons.

### 4.4. Effects of Cyanophyta on Plankton Community Structure

The Cyanophyta often inhibits the growth of aquatic organisms in eutrophic water [57]. However, in our study, according to the results of the linear regression, the Cyanophyta community showed a significant positive correlation with phytoplankton biomass and biodiversity index (Figure 6a,c). Cyanobacterial blooms reportedly occur when the Cyanophyta abundance exceeds 10^4^ cells/mL, which usually negatively affects the plankton communities and water quality [58]. In this study, Cyanophyta abundance did not meet the criteria at most sample sites in all seasons (Table 3). Referring to the methodology of Briland et al. [59], and in terms of the overall scope for the entire year, our results are consistent with the conclusion of Xu et al. [60], i.e., that plankton were not adversely impact at low Cyanophyta concentrations. Cyanophyta can promote phytoplankton diversity through nutrients release [61]. Despite studies showing that the zooplankton biodiversity does not respond well to the abundance of Cyanophyta, there was a significant positive correlation between Cyanophyta and zooplankton biomass (Figure 6b). The main reason is that most of the zooplankton species were comprised of rotifers and protozoans, which can feed on Cyanophyta without noticeable adverse effects when Cyanophyta abundance is low [62,63]. This was consistent with the findings of Zhao et al. [64] that rotifers usually dominate when Cyanophyta abundance is low. The zooplankton abundance was mainly dominated by protozoa and rotifers, with relatively low species richness and no significant seasonal variation in the biodiversity index (Figure 3d,e). As a result, Cyanophyta showed no significant impact on the zooplankton diversity index (Figure 6d,f).

## 5. Conclusions

(1) We elucidated the characteristics of the seasonal dynamics of plankton in the Yitong River and analyzed the effects of environmental factors and Cyanophyta on it. We identified 168 species of phytoplankton from nine phyla, mainly Cyanophyta, Chlorophyta, and Bacillariophyta. There are 93 species of zooplankton distributed among four phyla, mainly Rotifera and Protozoa.

(2) The phytoplankton community distribution in summer was less similar to those in spring and winter, and the environmental factors that significantly affected the phytoplankton community structure were WT, DO, and NO_3_^−^-N. There was no obvious seasonal difference in the distribution of the zooplankton community, and the influence of phytoplankton on zooplankton was more significant than that of environmental factors.

(3) Cyanophyta abundance was low at most sampling sites in the Yitong River and therefore it did not show a negative impact on the community structure of plankton in general. However, the abundance of Cyanophyta at the urban section of the river in summer has reached the criteria for bloom, further research should be carried out on the impact of Cyanophyta on the plankton community structure in the urban section of the Yitong River.

(4) With the rise of temperature, local river managers should control the discharge of nutrients to the Yitong River, and thus reduce the negative impact of temperature rise on water quality. Therefore, reducing the input of pollutants from N sources might be an effective way to reduce the eutrophication level of Yitong River.

## Figures and Tables

**Figure 1 ijerph-19-17030-f001:**
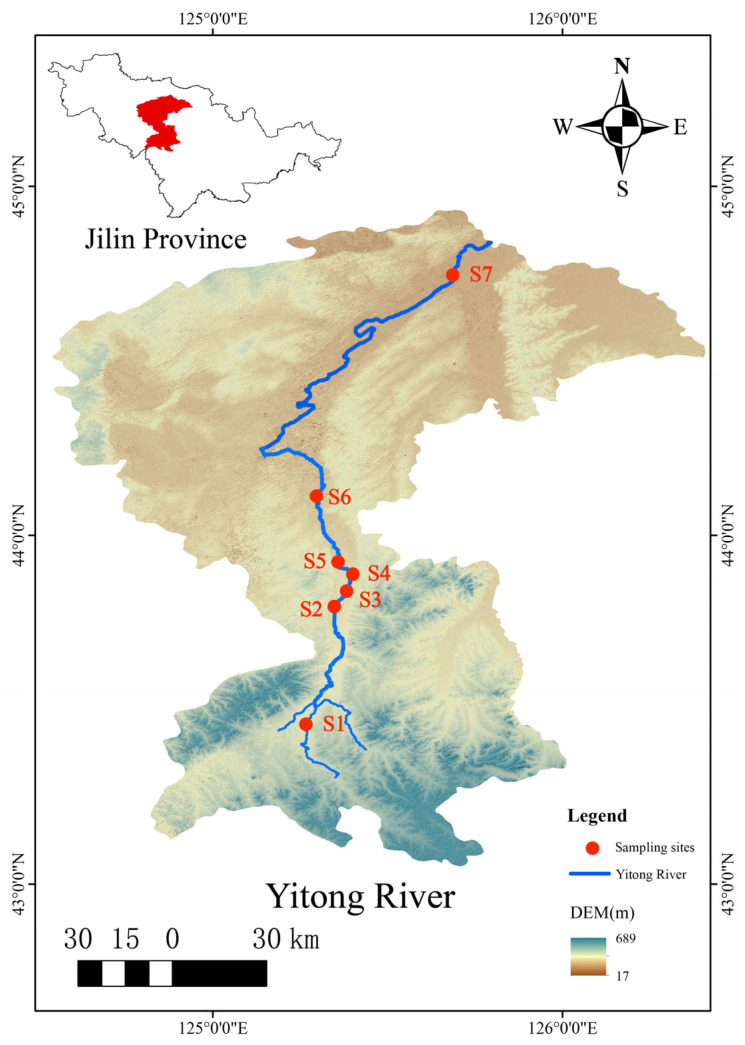
Location of the Yitong River and the sampling sites.

**Figure 2 ijerph-19-17030-f002:**
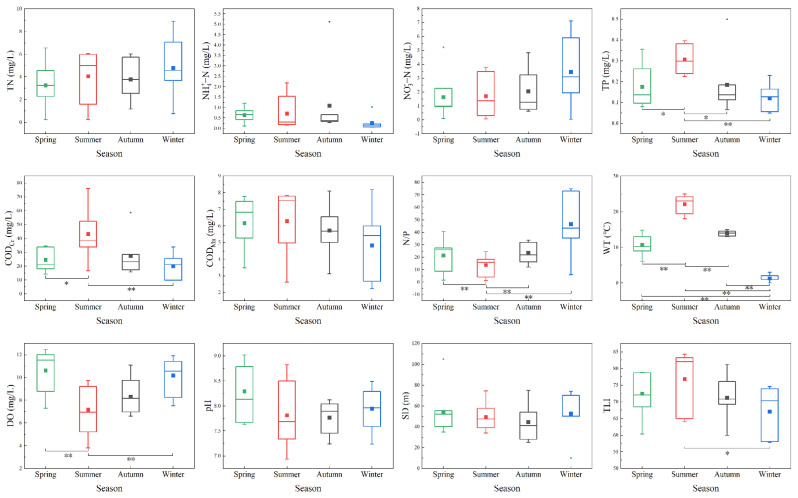
Seasonal variation of water quality indicators (* indicating significant differences at 0.05 level; ** indicating significant differences at 0.01 level). The three horizontal lines from top to bottom in the box plot represent the 75%, 50% and 25% quartiles, respectively. The vertical bar above the box represents the range of values from the 75% quartile to the maximum value, and the vertical bar below the box represents the range of values from the minimum value to the 25% quartile. Blocks of different colors indicate the average values.

**Figure 3 ijerph-19-17030-f003:**
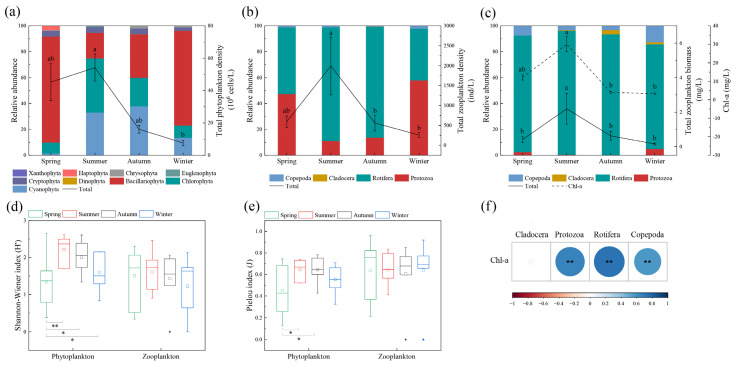
Seasonal variation of plankton abundance (**a**,**b**), biomass (**c**) and biodiversity (**d**,**e**), and the correlation between Chl-a and zooplankton biomass (**f**) (different letters on the error bars represent significant seasonal differences at 0.05 or 0.01 level in (**a**–**c**). * indicating significant differences at 0.05 level; ** indicating significant differences at 0.01 level in (**d**,**e**). ** indicating significant correlation at 0.01 level in (**f**)). Blocks of different colors in (**d**,**e**) indicate the average values.

**Figure 4 ijerph-19-17030-f004:**
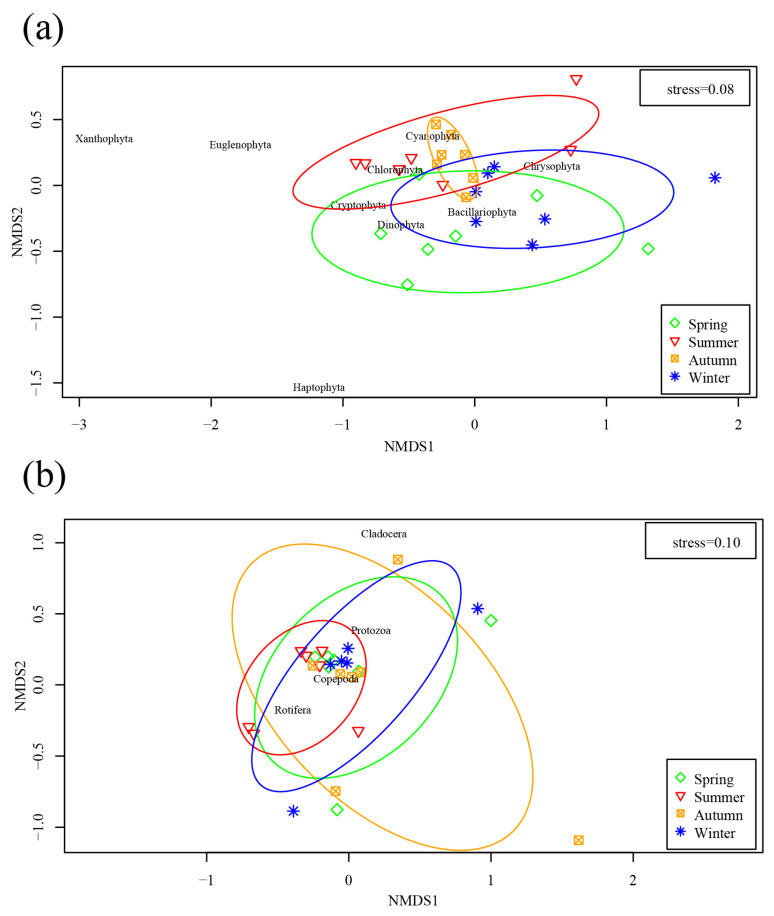
NMDS ordination plot of phytoplankton (**a**) and zooplankton (**b**) abundance.

**Figure 5 ijerph-19-17030-f005:**
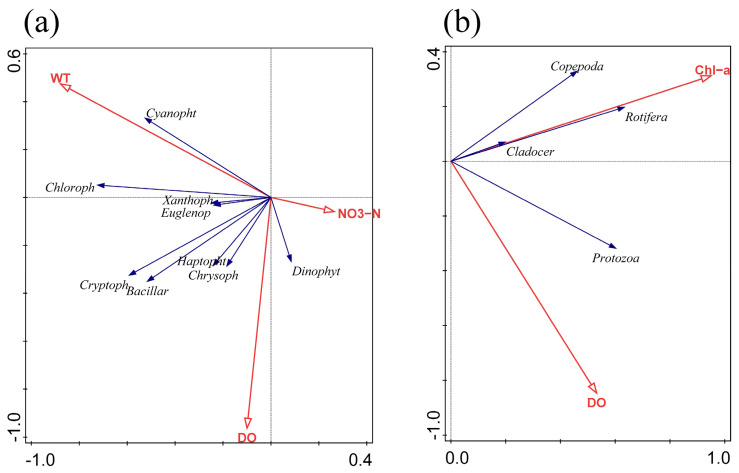
RDA of phytoplankton (**a**) and zooplankton (**b**) abundance with environmental factors.

**Figure 6 ijerph-19-17030-f006:**
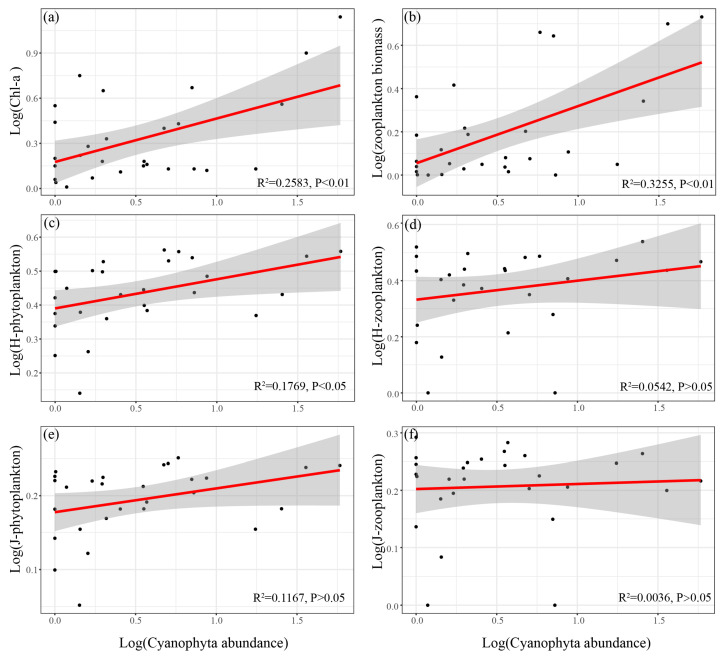
Linear relationship between log (Cyanophyta abundance), log (plankton biomass) (**a**,**b**), log (plankton biodiversity index) (**c**–**f**) (H-phytoplankton and H-zooplankton indicate the Shannon-Wiener index of phytoplankton and zooplankton; J-phytoplankton and J-zooplankton indicate the Pielou index of phytoplankton and zooplankton).

**Table 1 ijerph-19-17030-t001:** Sampling site characteristics of the Yitong River.

Sampling Sites	Abbreviations	Coordinates	Range of Depth	Date	Main Environmental Variables
Xingguang section	S1	125°14′ E, 43°26′ N	0.8–1.6 (m)	April, June, August, October and December	TN, NH_4_^+^-N, NO_3_^−^-N, TP, COD_Cr_, COD_Mn_, WT, DO, pH, SD, Q
Xinlicheng Reservoir dam section	S2	125°20′ E, 43°48′ N	0.2–0.8 (m)		
In front of the south gate of the South Third Ring Road	S3	125°21′ E, 43°49′ N	1.2–2.75 (m)		
In front of the Free Gate	S4	125°22′ E, 43°52′ N	1.3–2.15 (m)		
Yangjiawaizi Bridge section	S5	125°22′ E, 43°55′ N	1.6–2.8 (m)		
Baolong Bridge section	S6	125°16′ E, 44°5′ N	0.6–1.45 (m)		
Kaoshan Bridge section	S7	125°40′ E, 44°46′ N	0.7–1.3 (m)		

**Table 2 ijerph-19-17030-t002:** Significant environmental factors with RDA results.

Communities	Indicators	Explains	Contributions	*p*
Phytoplankton	WT	27.9%	42.4%	0.002
	DO	13.3%	20.1%	0.004
	NO_3_^−^-N	8.1%	12.3%	0.014
Zooplankton	Chl-a	33%	47.2%	0.002
	DO	10%	14.3%	0.01

**Table 3 ijerph-19-17030-t003:** The spatial and temporal distribution of Cyanophyta abundance (10^6^ cells/L) in the Yitong River.

	S1	S2	S3	S4	S5	S6	S7
Spring	3.73	0.00	0.00	0.42	0.00	0.43	0.00
Summer	0.70	1.54	24.36	34.89	57.26	0.98	6.05
Autumn	0.96	4.03	4.81	7.70	16.48	2.53	6.26
Winter	0.01	0.18	1.08	0.00	0.60	2.56	2.71

## Data Availability

Data available on request from the authors.

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
