# Peer review of "Seasonal Variation Characteristics and the Factors Affecting Plankton Community Structure in the Yitong River, China"

_ijerph, 2022, doi:10.3390/ijerph192417030_

Round 1
Reviewer 1 Report
There is one note to the work. The authors studied the plankton of the river. It is known that the flow velocity plays an important role in the formation of plankton communities in rivers. However, the article lacks information about this factor; it is not included in the statistical analysis. It is necessary to provide information on changes in current speed in different seasons, include this factor in the analysis or explain its absence. In the future, the authors need to take into account that rivers are characterized by a pronounced mosaic of areas with different environmental conditions, among which the main one is the flow velocity. In this regard, it is important to consider the similarities and differences in seasonal changes and interrelationships of plankton communities in areas with slow and fast currents, as well as under regulation conditions.Author Response
Please see the attachment.

Reviewer 2 Report
Review for the paper "Seasonal variation characteristics and the factors affecting plankton community structure in the Yitong River, China" by Ang Dong, Xiangfei Yu, Yong Yin, Ke Zhao submitted to "International Journal of Environmental Research and Public Health".
General comment.
Studies on seasonal patterns and dynamics in the freshwater environments are of great significance because they allow us to understand the phenology of crucial biotic components, among which phyto- and zooplankton are the most important. The authors investigated relationships between a set of environmental variables and plankton communities in the Yitong River, China in different seasons. They established clear temporal variations in major nutrients, hydrology as well as community structure of phytoplankton and zooplankton. The study expands our knowledge of the structure and functioning of plankton assemblages in riverine systems and may be interesting for scientists dealing with similar ecosystems. However, I feel the ms must be improved prior to being accepted for publication. In particular, comparisons with other regions are strongly needed to improve the ms.
Specific remarks.
L46, 51. Cyanophyta must be in ordinary font not Italic. According to the International nomenclature, the Latin names must be in Italic only for species and genera names. The same for L112, 116, 140, 176-177, 179, 181, 184-186, 187-191, 193, 199, 239-242, 250, 257, 260-261, 263, 266, 268-270, 273-274, 278-279, 306, 333-334, 339, 342-343, 347, 349-351, 355, 357-358, 360, 362, 364.
Introduction. Please, add some sentences regarding previous studies in the region.
L73-79. Highlight the novelty of your research. The aim of the study must be defined more clearly.
Section 2.1. I suggest the authors to provide a table showing characteristics of sampling sites (date, coordinates, depth, and main environmental variables).
Fig. 2 caption. Explain what do mean vertical bars in the plots.
L187-199. What about real estimations of zooplankton abundance and biomass? It would be good if seasonal measures expressed as individuals per L (abundance) and mg per L (biomass) be presented in the ms.
L192-199, The authors described the seasonal pattern of the zooplankton biomass but I have found no description of calculating this parameter. Provide a short section in M&M regarding procedures used to estimate zooplankton biomass.
Discussion. Please, compare your results with other similar ecosystems and regions focusing on similarities and dissimilarities between your study and other references. This would make your paper more interesting and attractive for international readers.
Round 2
Reviewer 2 Report
The authors have revised the paper according to my comments.